# Local pH oscillations witness autocatalytic self-organization of biomorphic nanostructures

M. Montalti[1], G. Zhang[2], D. Genovese[1], J. Morales[2], M. Kellermeier[3] & J.M. García-Ruiz[2]

Bottom-up self-assembly of simple molecular compounds is a prime pathway to complex materials with interesting structures and functions. Coupled reaction systems are known to spontaneously produce highly ordered patterns, so far observed in soft matter. Here we show that similar phenomena can occur during silica-carbonate crystallization, the emerging order being preserved. The resulting materials, called silica biomorphs, exhibit non-crystallographic curved morphologies and hierarchical textures, much reminiscent of structural principles found in natural biominerals. We have used a fluorescent chemosensor to probe local conditions during the growth of such self-organized nanostructures. We demonstrate that the pH oscillates in the local microenvironment near the growth front due to chemical coupling, which becomes manifest in the final mineralized architectures as intrinsic banding patterns with the same periodicity. A better understanding of dynamic autocatalytic crystallization processes in such simple model systems is key to the rational development of advanced materials and to unravel the mechanisms of biomineralization.

[1] Dipartimento di Chimica 'G. Ciamician', Alma Mater Studiorum, Università di Bologna, I-40126 Bologna, Italy. [2] Laboratorio de Estudios Cristalográficos, Instituto Andaluz de Ciencias de la Tierra (CSIC-Universidad de Granada), E-18100 Armilla (Granada), Spain. [3] Material Physics, BASF SE, RAA/OS—B007, D-67056 Ludwigshafen, Germany. Correspondence and requests for materials should be addressed to J.M.G.-R. (email: juanmanuel.garcia@csic.es).

living organisms are able to precisely control the formation of crystalline matter throughout their tissues, typically by using organic (macro)molecules that direct the assembly of the inorganic component into architectures with exceptionally complex shapes and textures[1–3]. Therefore, it is not surprising that most attempts to mimic biomineralization *in vitro* have relied on organic molecules[4] and investigated their effect on the nucleation and growth of prominent minerals like calcium carbonate[5] or phosphate[6]. By contrast, the inorganic-inorganic route for the formation of hierarchical structures has scarcely been explored. Yet, bizarre as it might appear, it has been shown that materials with features closely resembling those of biominerals can be obtained in simple single-pot experiments[7]. These so-called silica biomorphs form in alkaline media upon crystallization of carbonates like $BaCO_3$ or $CaCO_3$ under the influence of dissolved silicate species[8–10]. This process results in completely inorganic nanostructures that display (i) complex non-crystallographic morphologies with continuous curvature (such as regular helicoids, Fig. 1a) (refs 8–10); (ii) mesocrystal-type textures with long-range orientational order (Fig. 1b) (refs 9,11,12); (iii) banded patterns (Fig. 1c,d) (refs 7,13,14) and

(iv) structural features indicative of a growth mechanism involving the accretion of amorphous primary particles (inset in Fig. 1b) (ref. 15).

Given these striking analogies to natural biominerals and their relevance for material science[16] and early life detection[8,17], considerable effort has been made to understand the formation of silica biomorphs[7–10,14,18–21]. The current picture assumes a coupling of the chemical speciation of silicic and carbonic acid in solution, which drives the concerted precipitation of the two mineral components across all relevant length scales[9–10,19–21]. The model (schematically depicted in Fig. 1e,f) is based on the inverse pH dependence of the solubility of carbonate and silica: as carbonate crystals form at the front of a growing biomorph, the accompanying release of protons provokes a local decrease of pH and induces the precipitation of silica, which inhibits further carbonate growth. In turn, silica polycondensation causes an increase in pH and therefore triggers a new event of carbonate nucleation. In this way, the system maintains an autocatalytic cycle of co-precipitation, during which silica-coated carbonate nanocrystals are produced continuously and self-organize into two-dimensional surfaces that may later bend and twist to develop delicate helical or worm-like morphologies[9–10,20–21].

An immediate consequence of the envisaged growth mechanism is that the local pH near the active front should oscillate in time and space. Although there is evidence supporting the notion of local speciation changes[19,22], no direct proof for the oscillatory behaviour—and hence the autocatalytic nature of the process—has been achieved so far. The main difficulty for the detection of the predicted local variations in pH is the short length and time scales at which changes are expected to occur, and that absolute differences in pH are likely to be very small. In the present work, we have solved this problem by using fluorescence microscopy, a non-invasive *in situ* technique with high sensitivity[23,24]. The mapping of pH by fluorescence imaging requires the presence of a suitable pH-sensitive fluorescent chemosensor in the reaction medium. Here we have used acridine orange (Fig. 2a), a dye that gives almost no fluorescence signal in its neutral form (AO) prevailing at the typical bulk pH ($\sim$11.0) in our experiments, whereas a local decrease in pH will increase the concentration of the strongly fluorescent cationic form (AOH) (ref. 25). Consequently, the fluorescence signal at a given position provides direct information about the pH in the local microenvironment.

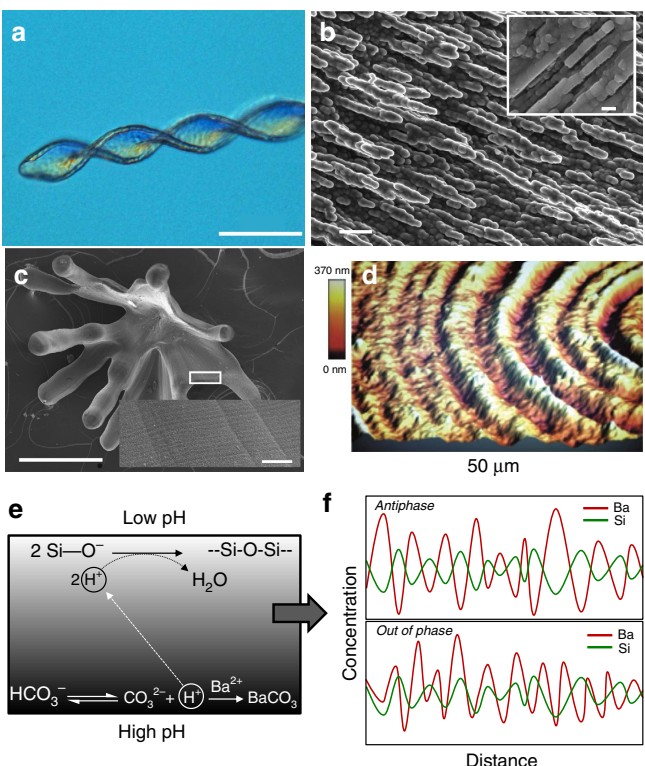

**Figure 1 | Complex self-assembly in purely inorganic systems.**
(**a**) Polarized optical micrograph of a barium carbonate biomorph with twisted morphology. (**b**) High-magnification SEM image revealing that biomorphs consist of numerous co-oriented carbonate nanocrystals (in this case aragonite). These nanorods grow by accretion of still smaller spherical subunits, as shown in the inset. (**c**) SEM image of a calcium carbonate biomorph (here monohydrocalcite) exhibiting periodic banding patterns (as seen in the magnified inset). (**d**) 3D reconstruction of an AFM height profile of a biomorph with sheet-like structure where periodic banding is clearly evident. (**e**) Scheme of the coupled chemical reactions leading to alternating precipitation of barium carbonate and silica during the formation of biomorphs. (**f**) Sketch of two different types of spatial composition banding that could arise from the model depicted in **e**. Scale bars: (**a**) 50 μm; (**b**) 1 μm; inset 100 nm; (**c**) 300 μm; inset, 10 μm.

## Results

**Mapping pH by fluorescence microscopy.** Figure 2b shows a sequence of fluorescence images collected from a biomorph aggregate that grew flat (two-dimensional sheet) and perpendicular to the optical axis of the microscope (see in the Fluorescence Microscopy section in the Supplementary Information for details on experimental methods and Movie 1 for the whole recorded time-lapse sequence of the growing structure). The bright fluorescence observed at the rim of the aggregate reveals enhanced protonation of the chemosensor and shows that the pH is indeed decreased in the proximity of the growth front relative to the surrounding alkaline bulk. In order to analyse the propagation of the fluorescence intensity in space, the time-lapse images were overlapped (Fig. 2c) and eight representative directions (D1-D8) along the growth vectors were selected to track the position of the fluorescence as well as its intensity. Figure 2d shows the intensity profiles along the growth vector D1 at different times, from which three important conclusions can be drawn. First, the position of the fluorescence peak moves linearly with time (Fig. 2g) at a rate of 0.21 μm min$^{-1}$, which is in a range typical for the growth of biomorphs[9,20] and thus confirms that the fluorescence peak

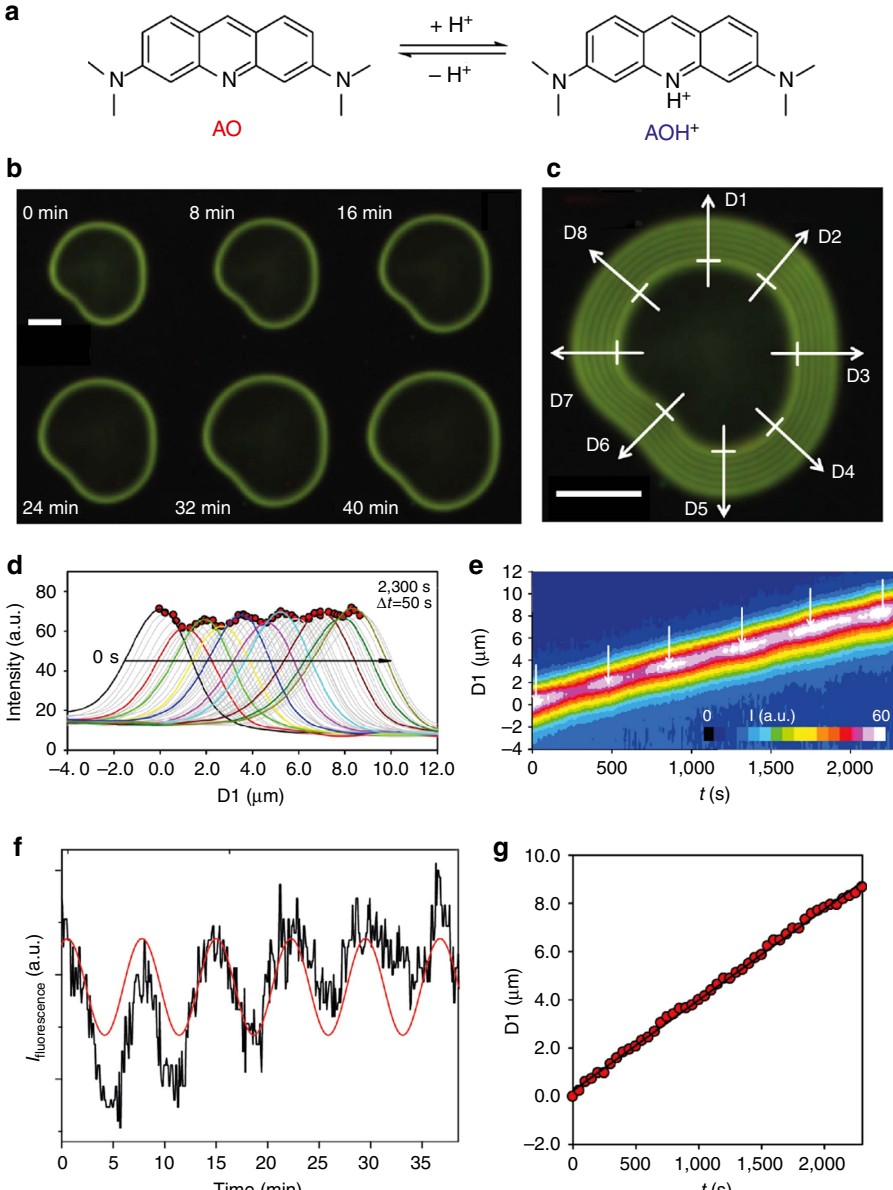

**Figure 2 | *In situ* measurements of local pH variations during the formation of silica biomorphs.** (**a**) Molecular structure of acridine orange in its neutral (AO) and protonated state (AOH$^+$). (**b**) Fluorescence images acquired at different times (as indicated) from a biomorph sheet growing in the presence of acridine orange (50 μM, $\lambda_{exc.} = 488$ nm). (**c**) Overlap of the shown images in **c** with eight selected directions along the respective growth vectors. (**d**) Fluorescence intensity profiles along the growth vector D1 at different times (50 s interval). Curves corresponding to relative intensity maxima and minima are plotted as thicker coloured lines. (**e**) Contour map showing the evolution of the fluorescence intensity along vector D1 with time. Arrows point to the periodically observed maxima. (**f**) Plot of the absolute intensity at the fluorescence peak maxima in **d** as a function of time. The red line is a sine function fitted to the experimental data. (**g**) Time-dependent change of the position of the fluorescence peak maxima in **d**. The black line is a linear fit of the data, from which the rate of advance of the reaction front can be obtained. Both the trend and the absolute rate values agree well with data reported for silica biomorphs in the literature[9,20]. Scale bars: 10 μm.

advances along with the growth front. Second, the peak shape and width do not change significantly with time, demonstrating that the pH variation occurs within a confined region extending ca. 2 μm beyond the growing front, in good agreement with indirect estimations[19] and recent experimental measurements[22]. The third and most important observation is that the absolute value of the fluorescence intensity at the peak maximum is not constant over time, but clearly shows oscillatory behaviour. This becomes evident when the intensity profiles of all 467 collected frames are combined together in the form of a contour map (Fig. 2e), or when plotting the absolute intensity at the fluorescence peak maximum as a function of time (Fig. 2f). During the monitored

period (2,200 s), five complete oscillations (six maxima) can be observed over a total distance of 8.1 μm, giving an average peak-to-peak separation of 1.5 μm and an oscillation frequency of 0.14 min$^{-1}$ (see the SI for corresponding analyses along the vectors D2-D8).

These results provide clear evidence for the existence of local pH oscillations at the growth front of silica biomorphs. Calibrations of the measured fluorescence intensities (see Fluorescence microscopy section in the Supplementary Information) suggest that the average pH in the local micro-environment is about 9.7 (that is, significantly lower than in the bulk, as expected), while the amplitude of the pH oscillation was

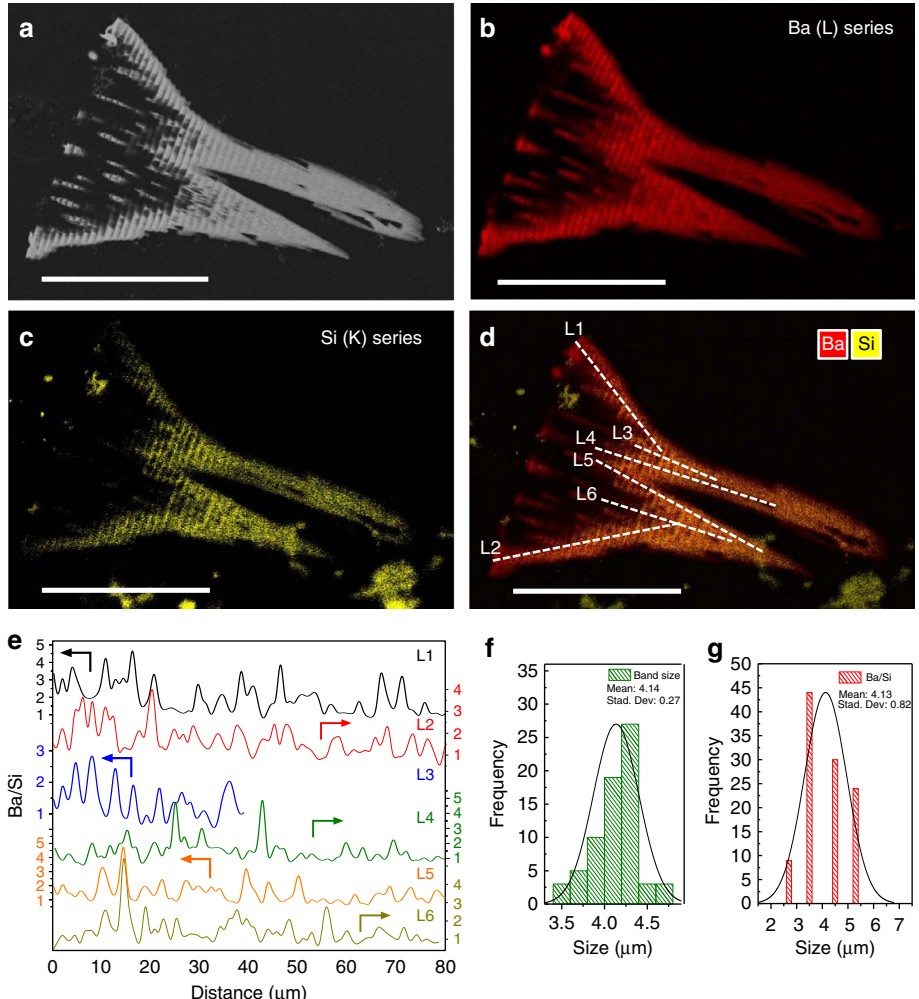

**Figure 3 | Chemical composition oscillations in silica biomorphs.** (**a**) Backscattered electron image taken from a thin section of a biomorph with trumpet-like morphology. (**b**) EDX mapping of the Ba distribution. (**c**) EDX mapping of the Si distribution. (**d**) Superposition of the Ba (red) and Si (yellow) signals. White lines represent the paths along which the composition was analysed. (**e**) Variation of the Ba/Si ratio along the different lines in **d**. (**f**) Histogram of the width of the structural banding observed in **a**. (**g**) Histogram of the width of the compositional banding observed in **e**. Scale bars are 100 μm.

estimated to be ca. 0.2 units relative to the average value. In terms of supersaturation, this difference of 0.2 units of pH means a saturation index of 1.26 for silica and 1.44 for carbonate, assuming that system is closed. If open to the air, that value increases to 2.5. These values are large enough to trigger nucleation and growth of slightly soluble compounds such as $BaCO_3$ and $SiO_2$. The precise periodicity and scale of the oscillation wavelength exclude any external perturbation as possible reason for the observed trends, and rather imply that the detected periodic variations are intrinsic to the growth process, as also suggested by recent experiments in single-phase systems[26].

**Analysis of self-assembled banding structures**. In principle, the observed oscillatory behaviour could be the result of either classical reaction-diffusion coupling (like for example in Liesegang rings)[7,27], or reaction-reaction coupling as shown in Fig. 1e. While rhythmic banding patterns would be expected in both cases, the latter scenario should in addition lead to periodic changes in composition, as illustrated in Fig. 1f. In order to address this issue, we have performed *ex-situ* analyses on biomorphs grown with the same protocol as in the fluorescence

microscopy experiments. Atomic force microscopy (AFM) images of sheet-like aggregates reveal a wavy topography with periodic variations in thickness on a length scale ranging from 1 to 6 μm (Fig. 1d), in good agreement with previous studies on biomorphs of barium carbonate[14,26] and monohydrocalcite[28]. In all these cases, the periodicity of the banding patterns matches well the frequency of the pH oscillation observed by fluorescence microscopy, and moreover the banding does not show the square-root dependence on time typical for reaction-diffusion systems[27]. Most probably, thin regions are formed in phases where the local pH is low (depressing carbonate crystallization and enhancing silica precipitation), while thicker parts correspond to higher pH values in the microenvironment.

So far, spatial oscillations in chemical composition as expected for this scenario have not been observed for intact biomorphs[14]. Here, we have probed the composition after polishing the structures, in order to get access to their internal volume and thus to avoid interferences of the outer silica skin that usually covers the biomorphs[20]. Elemental mapping of the inner part of the aggregates shows a clear picture (Fig. 3): the backscattered electron image (Fig. 3a) as well as the signals for both Ba (Fig. 3b) and Si (Fig. 3c) display periodic variations along the respective growth direction. More importantly, this periodicity is still

evident in profiles of the Ba/Si atomic ratio along different chosen growth vectors (Fig. 3d,e), demonstrating that the local composition in fact oscillates around some average value. In most regions, the concentration of barium is significantly higher than that of silica (usually by a factor of up to 5), indicating that $BaCO_3$ precipitation dominates the process, in line with previous observations[14,20,22]. Statistical data analysis confirms that the mean width of the observed bands is about 4 μm (Fig. 3f) and that the oscillation in composition has the same frequency (Fig. 3g), both again well in line with the local variations in pH traced *in situ* during growth (Fig. 2f). When analysing these composition profiles in more detail, it appears that the oscillations in silica and barium content are not exactly in antiphase (as anticipated in Fig. 1f), but clearly they are out of phase. This shows that the mineralization of the two components is not independent, likely because carbonate and silicate influence each other through their solution speciation as depicted in Fig. 1e.

## Discussion

Our results show that the coupling of carbonate and silicate precipitation induces periodic behaviour at the growth front, which leads to oscillations in local conditions that are directly correlated with topographic and compositional banding patterns found in the final mineralized structures. Autocatalytic processes like those described here are well established in the field of solution chemistry, where the coupling of simple reactions can lead to temporary out-of-equilibrium patterns as in the case of the famous Belousov-Zhabotinsky systems[29]. It has also been reported that the order created in this way can be preserved in the form of 2D concentric ring structures when one of the involved components mineralizes in the course of the process[30]. The present study shows that reaction coupling of two co-precipitating phases is able to drive the self-organization of much more complex 3D architectures that display structural features otherwise only known from biologically controlled crystallization. Subtle changes in local conditions have proven to be key to the bottom-up assembly of elaborate hierarchical textures, and it may well be envisaged that related effects play a much greater role in natural biomineralization than hitherto believed.

## Methods

**Growth of silica biomorphs.** The synthesis of silica-barium carbonate biomorphs was performed in aqueous solution following previously established protocols[5,20,21]. First, 1 ml of commercial water glass (Sigma-Aldrich, reagent grade, Ref. 338443) was diluted in 350 ml water. The resulting solution was mixed with a small amount of 0.1 M NaOH to set the pH to approximately 11. Then an equal volume of 0.01 M $BaCl_2$ solution was mixed in the diluted water glass solution, which results in a final mixture with a pH value of 10.6 ± 0.2 that contains 5 mM Ba and 7.5 mM $SiO_2$. Finally, 10 ml of this mixture were combined with 100 μl of a 1 mM solution of acridine orange (Sigma-Aldrich) in water.

Crystallization of barium carbonate was triggered by the diffusion of atmospheric carbon dioxide into the above-described alkaline solution[21]. In order to minimize background fluorescence and thus maximize spatial resolution, the experiments were performed in thin crystallization cells with a thickness of ~100 μm as shown in Supplementary Fig. 1: a thin layer of growth solution was sandwiched between a glass slide and a coverslip that were separated by a spacer with two open holes for the uptake of $CO_2$. In particular, two strips of double-sided Scotch tape (5 × 40 mm) were attached parallel and close to the two longer edges of a microscopy slide (75 × 25 mm) at a distance of 15 mm from each other. Then 100 μl of growth solution was dropped into the centre and a glass coverslip (40 × 25 mm) was placed on the free side of the tape to close the cell. This configuration allowed for an optimized spatial resolution in the fluorescence signal, as the thickness of the layer of bulk solution illuminated during the experiment and hence the background emission was reduced (without any noticeable interference with the growth of biomorphs). In addition, the use of thin growth cells minimizes convective flow so that the biomorphs grow with little perturbation at a given location, facilitating measurements of their growth rate.

**Fluorescence microscopy.** UV-Vis absorption and fluorescence measurements were performed using a PerkinElmer Lambda 25 UV-Vis spectrophotometer and a FluoroMax-4 (HORIBA) spectrofluorimeter respectively. The growth of the biomorphs was monitored *in situ* by wide-field fluorescence microscopy (Olympus IX-71 inverted microscope) under low-power excitation conditions. The wide-field illumination for excitation was achieved by focusing the expanded and collimated laser beam (Argon laser, Melles-Griot 534 Series, $\lambda_{exc}$ = 488 nm) onto the back-focal plane of the objective (UPLFLN10X2, Köhler illumination mode) using a suitable dichroic mirror (Chrome 500dcxr). In this experimental setup, the dichroic mirror irradiated the specimen with the desired wavelength and then separated the much weaker emitted light (that is, the fluorescence) from the excitation light. After being further filtered, the emitted light reached the eye or the detector so that the fluorescent areas were contrasted against a dark background. The detection limit is largely determined by the darkness of the background, of which the emitted light should typically be $10^5$ or $10^6$ times weaker than the excitation light. Fluorescent images were collected using a CCD camera (Basler Scout scA 640–70) and further converted into videos using ImageJ to visualize the growth of biomorphs with fluorescence contrast. Videos were recorded during the first 12 h of the experiments, with a typical interval between frames of 5 s.

In order to convert the measured intensity data into actual (local) pH values, a calibration curve was acquired using $10^{-5}$ M solutions of AO in water, which were adjusted to different pH levels by adding small volumes of either HCl or NaOH. Fluorescence intensity was measured with the same spectrofluorimeter as used for the growth experiments, while pH was monitored in parallel with a Jenway 4330 pH-meter equipped with a Crison sensor. The resulting dependence of the fluorescence intensity $I$ on pH is shown in Supplementary Fig. 3. The experimental data were fitted with the following equation:

$$I = I_2 + \frac{I_1 - I_2}{1 + 10^{-(pk_a - pH)}} \qquad (1)$$

where $I_1$ and $I_2$ are the intensities measured for AO solutions at pH 7.0 and 13.0, respectively. This gave a pKa value of 10.0 ± 0.1 for the AO dye. Using this calibration curve, the fluorescence intensities measured in the growth experiments were converted into pH values according to:

$$pH = pk_a + \log\left(\frac{I_1 - I}{I - I_2}\right) \qquad (2)$$

The spacial resolution of the fluorescence intensity measurements by microscopy was 0.9 μm for the experimental set up. The fluorescence images shown in Fig. 2b reveal that a small fraction of the AO dye is incorporated in the biomorph during growth. Considering the fluorescence intensity in the inner part of the crystal aggregates, where growth is terminated, we could estimate the concentration of the dye in the biomorph to be comparable with that in the silicate solution (i.e. $10^{-5}$ M) and hence the content of AO in the biomorphs was less than 3 p.p.m.. Such low concentrations can hardly affect precipitation, and indeed we observed very similar growth rates values, growth rate trend, shapes and sizes of biomorphs with and without the dye. For instance, the growth of biomorphs in our fluorescence experiments was found to be linear in time and the growth rate to vary in the order of tens of microns per hour (Fig. 2e), in good agreement with previous experiments. In addition, the wavelength of the pH oscillations measured in our fluorescence experiments (1.5 microns) is similar to the wavelength of textural and compositional banding (4.1 microns) in the final structures obtained in this work (Fig. 3f,g) as well as in previous studies on MHC biomorphs[28] and $BaCO_3$ biomorphs[15,26] (that is, in the range between 1 and 6 microns).

**Ex-situ characterization.** Biomorphs grown according to the above-described protocol (with or without the fluorescent dye) were isolated by removing the mother solution after predefined times, followed by repeated rinsing with water and ultimately ethanol. Optical micrographs of the resulting structures were taken using a Nikon AZ100 microscope. For AFM, small glass slides were added to the growth solution as substrates. Biomorphs formed thereon remained stuck even upon extensive rinsing and thus they could be directly investigated after drying. AFM studies were performed on an Autoprobe CP microscope (Park Scientific Instruments) in non-contact mode using cantilevers (non-contact ultralevers, ULCN-AUMT-A) with a resonance frequency of 80 kHz. Finally, for scanning electron microscopy (SEM) and elemental analyses by energy-dispersive X-ray (EDX) spectroscopy, suitable biomorphic structures were selected under a stereomicroscope and embedded within epoxy resin using a glass slide as substrate. The resulting specimen was sequentially polished with diamond powder up to a grain size of 1 μm until the internal structure of the biomorphs was exposed at the surface. The sample was eventually coated with carbon and studied by SEM using a Zeiss SUPRA 40 VP microscope equipped with an X-Max 50 mm EDX system from Oxford Instruments.

**Data availability.** The authors declare that the data supporting the findings of this study are available within the paper and its Supplementary Information files.

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

## Acknowledgements

This work was funded by the European Research Council (European Union's Seventh Framework Programme (FP7/2007-2013, grant no. 340863) and the Spanish MINECO grant CGL2010-16882 (co-funded by FEDER). We thank Julian Opel for help with some of the experiments and Isabel Guerra-Tschuschke for technical assistance with electron microscopy.

## Author contributions

M.M., G.Z. and D.G. performed the fluorescence microscopy experiments, analysed the data and discussed the results. J.M. performed the electron microscopy studies and analysed the data. M.K. performed the AFM studies, analysed the data and wrote the paper. M.M. and J.M.G-R. designed the experiments. J.M.G-R. conceived the idea, discussed the results and wrote the paper.

## Additional information

**Competing financial interests:** The authors declare no competing financial interests.

**Publisher's note**: 

