## [Peer Review File · Nature Communications]

Reviewers' Comments:

Reviewer #1 (Remarks to the Author)

Local pH Oscillations Witness Autocatalytic Self-Organization of Biomorphic Nanostructures.

I have reviewed this paper for another journal in an earlier form. Below I update my comments based on the new manuscript. Notably the authors have estimated the pH shift, which is nice addition to the paper.

This is a well written paper and a beautiful experiment that I believe will be of interest to Nature Communications readers. The paper's greatest strength/novelty is the use of the fluorescence to measuring the pH oscillations at the growth front. The direct measurement and quantification of pH sets it apart from other experiments. From a crystal growth perspective, the greatest weakness is the lack of quantification of the supersaturation and the impact of the pH shift on growth rates.

It may be that pH dependent supersaturations and growth rates are not available in the literature making quantification outside the scope of this paper. If so, many of my concerns cannot be addressed easily. In this case I find figure 1f to be misleading but otherwise have no major objections.

Concerns:

1) Fluorophore Control

How do we know that the fluorophore does not actively participate/alter barium carbonate or silica growth? Likely the authors have tested this but it this control is not described in the supplemental materials. For example, is the oscillation period and Ba/Si ratio the same? Is there any evidence of fluorophore inclusion in the bimorph (as measured by a spectroscopic technique).

2) pH quantitation and its repercussions on supersaturation and composition

The authors visualize and estimate the pH shift at the growth interface. The work would be stronger if the impact of this pH shift on supersaturation (or growth rate) were discussed. Does a pH shift of 0.2 change the supersaturation or growth rate of Silica and Barium carbonate by a little or a lot? The authors seem to have a quantitative model in mind as shown in 1f. This model

is not described fully enough to review (for example is this meant to reflect kinetics or thermodynamics).

The speciation model in Figure 1F represents a certain pH shift but this value is not given anywhere. Is the experimentally measured pH shift (and Ba, Si concentrations) being modeled? Also the Fig1e suggests that the pH shift leads to changes in BaCO₃ and silica growth rates but those values are not provided. Nevertheless, the modeled and measured Ba/Si composition variations shown in Figure 1F/3e are similar - is this warranted by known pH dependent growth rates? By showing each piece the paper tacitly suggests that these pieces are in agreement when not enough information is provided to judge whether they should be linked.

Comments

Figure 1D caption – a sheetlike structure of what?

Figure 1F Not enough information is provided to critically evaluate. Requires more description of the model and model inputs. Have the authors used the concentrations and pH values used in the experiment? Are the composition variations based on measured growth rates or theoretical supersaturation or is this purely schematic with the xy-axis values added to mimic the variations found via EDX? If the axis values are ad hoc then state this explicitly.

Figure 2G and corresponding text: The growth rate is more meaningful with the supersaturation.

Supplemental - Growth of Silica biomorph: A good description of the growth protocol is provided. This could be made better by including the final molarities of all the chemicals (for example, the final molarities were X sodium silicate, 5mM BaCl₂ and 0.01 mM acridine,) as would be needed to estimate supersaturations, [Ba]/[Si] ratio, [acridine]/[Ba] ratio etc. – these are the quantities that effect growth rates and impurity interactions. For example sodium silicate is given in v/v instead of molarity. Ideally the supersaturation ranges for each species would be given. Although the supersaturations of BaCO₃ and silicate are drifting due to the interaction with CO₂, the supersaturation at pH 10.6 and 9.7 +/- 0.2 are particularly relevant for this paper.

Reviewer #2 (Remarks to the Author)

In the present work, using fluorescent chemosensor the authors probed locally pH oscillations during the growth of such biomorph for the first time, therefore demonstrating experimentally that the model they proposed previously for biomorphs growth was correct. This is a nice contribution that open new avenues for thinking in the field of biomineralization.

Dear Editor,

We acknowledge very much the comments, criticisms and suggestions made by the referees. We have studied them carefully and we are sending you below a point-by-point response (in blue) to these comments along with the changes (underlined, bold blue) that we have made in the text of the article and in the supplementary information in order to enhance the manuscript.

Referee 1:

This is a well written paper and a beautiful experiment that I believe will be of interest to Nature Communications readers. The paper's greatest strength/novelty is the use of the fluorescence to measuring the pH oscillations at the growth front. The direct measurement and quantification of pH sets it apart from other experiments. From a crystal growth perspective, the greatest weakness is the lack of quantification of the supersaturation and the impact of the pH shift on growth rates.

It may be that pH dependent supersaturations and growth rates are not available in the literature making quantification outside the scope of this paper. If so, many of my concerns cannot be addressed easily. In this case I find figure 1f to be misleading but otherwise have no major objections.

We thank very much the referee for her/his words.

As the referee suspect, there is not available literature on the growth rate versus supersaturation for witherite. Nevertheless, we have calculated and included in the revised version the supersaturation values for the amplitude of the measured pH oscillations (see below).

Concerns:

1) Fluorophore Control

How do we know that the fluorophore does not actively participate/alter barium carbonate or silica growth? Likely the authors have tested this but it this control is not described in the supplemental materials. For example, is the oscillation period and Ba/Si ratio the same? Is there any evidence of fluorophore inclusion in the biomorph (as measured by a spectroscopic technique).

We have estimate the concentration of fluorophore in the biomorph to be similar to the concentration in the bulk, roughly 10^{-5} M. This low concentration can hardly affect precipitation, and indeed we observe very similar growth rates values, growth rate trend, shape and size of the biomorphs with and without the dye. For instance, as shown in Figure 2E,

the growth rate of biomorphs in our fluorescence experiments has been shown to be linear with time and to vary in the order of tens of microns per hour, in good agreement with the trend and value found in many other experiments. In addition, the wavelength of the pH oscillations measured in our fluorescence experiments (1.5 microns) is also similar to the wavelength of textural and compositional banding (4.1 microns) measured in the experiments (Figure 3f and g) as well as those measured for MHC biomorphs (reference 28 in our paper) and BaCO₃ biomorphs (references 15 and 26 in our paper), i.e. in the range between 1 and 6 microns.

We have included in the supplementary information the following text including the concentration of fluorescence in the biomorphs and how we have calculate it:

The space resolution of the fluorescence intensity measurements by optical microscopy is 0.9 μm for the experimental set up. Fluorescence images shown in figure 2B reveal that a small fraction of the dye AO is incorporated in the biomorph during its growth. Considering the fluorescence intensity in the inner part of the crystal, where the growth is terminated, we could estimate the concentration of the dye in the biomorph to be comparable with the one of the silicate solution (10⁻⁵ M) and hence the content of AO to be as low as less than 3 ppm. This low concentration can hardly affect precipitation, and indeed we observe very similar growth rates values, growth rate trend, shape and size of the biomorphs with and without the dye. For instance, the growth rate of biomorphs in our fluorescence experiments has been shown to be linear with time and to vary in the order of tens of microns per hour (Figure 2E), in good agreement with the trend and value found in many other experiments. In addition, the wavelength of the pH oscillations measured in our fluorescence experiments (1.5 microns) is also similar to the wavelength of textural and compositional banding (4.1 microns) measured in the experiments (Figure 3f and g) as well as those measured for MHC biomorphs²⁸ and BaCO₃ biomorphs^{15,26}, i.e. in the range between 1 and 6 microns.

2) pH quantitation and its repercussions on supersaturation and composition

The authors visualize and estimate the pH shift at the growth interface. The work would be stronger if the impact of this pH shift on supersaturation (or growth rate) were discussed.

Our fluorescence experiments are at the frontier of what is today

technically achievable in terms of time and space resolution. We have designed the experiments to obtain fluorescence images at the highest possible resolution to reveal the existence or not of oscillatory waves of pH value. The cost we have paid is not to have at the same time the real image of the experiments and therefore we have not able to measure precise values of growth rate. We are working on a new experimental design to have both types of images recorded simultaneously in the future.

Does a pH shift of 0.2 change the supersaturation or growth rate of silica and barium carbonate by a little or a lot? The authors seem to have a quantitative model in mind as shown in 1f. This model is not described fully enough to review (for example is this meant to reflect kinetics or thermodynamics).

The pH shift of 0.2 units is clearly large enough to precipitate silica and carbonate. The supersaturation achieved by that drop (or increase) of pH depends on pH. Our experimental results show that the pH at the growth front is 9.7. For a drop of pH from 9.8 to 9.6 the fall in concentration is from $[C_{9.8}] = 255 \text{ mg/L}$ to $[C_{9.6}] = 206 \text{ mg/L}$, i.e. a supersaturation value of 1,24. For the case of BaCO_3 for the same drop of pH we obtain that the saturation index from pH 9.8 to pH 9.6 is 1,44 considering the system to be a closed system. If the system is open to the air, then the supersaturation value created by the oscillation of pH is 2.51. These values are large enough to trigger nucleation and growth of a slightly soluble compound such as BaCO_3 .

We have inserted this information in the text as follows:

In terms of supersaturation, this difference of 0.2 units of pH induces a saturation index of 1.26 for silica and 1,44 for carbonate assuming the system closed. If open to the air, that values increases to 2.5. These are values large enough to trigger nucleation and growth of slightly soluble compounds such as BaCO_3 and SiO_2 .

The speciation model in Figure 1F represents a certain pH shift but this value is not given anywhere. Is the experimentally measured pH shift (and Ba, Si concentrations) being modeled? Also the Fig1e suggests that the pH shift leads to changes in BaCO_3 and silica growth rates but those values are not provided. Nevertheless, the modelled and measured Ba/Si composition variations shown in Figure 1F/3e are similar - is this warranted by known pH dependent growth rates? By showing each piece the paper tacitly suggests that these pieces are in agreement when not enough information is provided to judge whether they should be linked.

The aim of Figure 1f, which appears in the introduction of the paper, is

not to provide a model of the oscillatory behaviour but to make easier for the reader the understanding of the problem we are trying to solve. As described in the caption of the figure, this is an “Sketch of the oscillations of composition in space”. To underline that this figure is not a result but a visual output of the hypothesis we are trying to test, we have shown two different plausible outputs of the oscillatory behavior of pH. We think that now Figure 1f is clearer and we thank the referee for his/her criticism. Consequently we have modified the caption of the Figure to say:

(f) Two different type of spatial composition banding that could arise from the model depicted in (e).

Comments

Figure 1D caption – a sheetlike structure of what?

Thank you for calling the attention on this. We have modified the text to say:

“3D reconstruction of an AFM height profile of a biomorph with sheet-like structure where periodic banding is also clearly evident”

Figure 1F Not enough information is provided to critically evaluate. Requires more description of the model and model inputs. Have the authors used the concentrations and pH values used in the experiment? Are the composition variations based on measured growth rates or theoretical supersaturation or is this purely schematic with the xy-axis values added to mimic the variations found via EDX? If the axis values are ad hoc then state this explicitly.

Thank you for this input. As written in its caption, Figure 1F is a sketch that aims to facilitate the reader the understanding that the chemical coupling of silica and carbonate precipitation must induce a oscillatory behavior of pH that could also be expressed by a compositional zoning. As written above, we have modified the sketch to show two possible types of zoning and make clear that the y axis values are ad hoc.

Figure 2G and corresponding text: The growth rate is more meaningful with the supersaturation.

We understand very well the suggestion of the referee, as in the classical literature on crystal growth this is the case. However, note that the main aim of this figure is to provide the detailed position of the fluorescence peak maxima. This plot also shows that the growth rate is constant in time. More rigorously, what can be measured from these data is the rate

of advance of the fluorescence maxima. Consequently we have changed the text in the caption to say:

“The black line is a linear fit of the data, from which the rate of advance of the reaction front can be obtained. Both the trend and the rates values fit the data of growth rate found in silica biomorphs”.

Supplemental - Growth of Silica biomorph: A good description of the growth protocol is provided. This could be made better by including the final molarities of all the chemicals (for example, the final molarities were X sodium silicate, 5mM BaCl₂ and 0.01 mM acridine,) as would be needed to estimate supersaturations, [Ba]/[Si] ratio, [acridine]/[Ba] ratio etc. – these are the quantities that effect growth rates and impurity interactions. For example sodium silicate is given in v/v instead of molarity. Ideally the supersaturation ranges for each species would be given. Although the supersaturations of BaCO₃ and silicate are drifting due to the interaction with CO₂, the supersaturation at pH 10.6 and 9.7 ± 0.2 are particularly relevant for this paper.

Thank you very much for these insightful suggestions. We have now expressed the sodium silicate concentration in molarity, actually the SiO₂ concentration, which is actually what matter. Actually we have written an even more detailed description of the preparation of biomorphs:

First, 1 mL of commercial water glass (Sigma-Aldrich, reagent grade, Ref. 338443) was diluted in 350 mL water. The resulting solution was mixed with a small amount of 0.1 M NaOH to set the pH to approximately 11. Then an equal volume of 0.01 M BaCl₂ solution was mixed the diluted water glass solution, which results in a final mixture with a pH value of 10.6 ± 0.2, that contains 5 mM Ba and 7.5 mM SiO₂.

The relevant values of carbonate supersaturation at pH 9.6 and 9.8 have been also inserted in the text (see above)

Referee 2

In the present work, using fluorescent chemosensor the authors probed locally pH oscillations during the growth of such biomorph for the first time, therefore demonstrating experimentally that the model they proposed previously for biomorphs growth was correct. This is a nice contribution that open new avenues for thinking in the field of biomineralization.

We acknowledge very much the kind words of the referee and his/her evaluation of the merit of our work. We also believe that silica biomorphs are fascinating structures that are helping to understand biomineralization in addition to inspire new routes for biomimetic materials and self-organization.

Reviewers' Comments:

Reviewer #1 (Remarks to the Author):

The authors have addressed my concerns.